# Feeding Interventions for Infants with Growth Failure in the First Six Months of Life: A Systematic Review

**DOI:** 10.3390/nu12072044

**Published:** 2020-07-09

**Authors:** Ritu Rana, Marie McGrath, Paridhi Gupta, Ekta Thakur, Marko Kerac

**Affiliations:** 1Indian Institute of Public Health Gandhinagar, Gujarat 382042, India; drparidhigupta@gmail.com (P.G.); ekta17020@rediffmail.com (E.T.); 2Department of Population Health, London School of Hygiene & Tropical Medicine, London WC1E 7HT, UK; marko.kerac@lshtm.ac.uk; 3Nutrition Research Advisor (MAMI), GOAL Global, A96 C7W7 Dublin, Ireland; 4Emergency Nutrition Network, Oxford OX5 2DN, UK; marie@ennonline.net; 5Centre for Maternal, Adolescent, Reproductive & Child Health (MARCH), London School of Hygiene & Tropical Medicine, London WC1E 7HT, UK

**Keywords:** infant, growth failure, malnutrition, breastfeeding

## Abstract

(1) Introduction: Current evidence on managing infants under six months with growth failure or other nutrition-related risk is sparse and low quality. This review aims to inform research priorities to fill this evidence gap, focusing on breastfeeding practices. (2) Methods: We searched PubMed, CINAHL Plus, and Cochrane Library for studies on feeding interventions that aim to restore or improve the volume or quality of breastmilk and breastfeeding when breastfeeding practices are sub-optimal or prematurely stopped. We included studies from both low- and middle-income countries and high-income countries. (3) Results: Forty-seven studies met the inclusion criteria. Most were from high-income countries (*n* = 35, 74.5%) and included infants who were at risk of growth failure at birth (preterm infants/small for gestational age) and newborns with early growth faltering. Interventions included formula fortification or supplementation (*n* = 31, 66%), enteral feeds (*n* = 8, 17%), cup feeding (*n* = 2, 4.2%), and other (*n* = 6, 12.8%). Outcomes included anthropometric change (*n* = 40, 85.1%), reported feeding practices (*n* = 16, 34%), morbidity (*n* = 11, 23.4%), and mortality (*n* = 5, 10.6%). Of 31 studies that assessed formula fortification or supplementation, 30 reported anthropometric changes (*n* = 17 no effect, *n* = 9 positive, *n* = 4 mixed), seven morbidity (*n* = 3 no effect, *n* = 2 positive, *n* = 2 negative), five feeding (*n* = 2 positive, *n* = 2 no effect, *n* = 1 negative), and four mortality (*n* = 3 no effect, *n* = 1 negative). Of eight studies that assessed enteral feed interventions, seven reported anthropometric changes (*n* = 4 positive, *n* = 3 no effect), five feeding practices (*n* = 2 positive, *n* = 2 no effect, *n* = 1 negative), four morbidity (*n* = 4 no effect), and one reported mortality (*n* = 1 no effect). Overall, interventions with positive effects on feeding practices were cup feeding compared to bottle-feeding among preterm; nasogastric tube feed compared to bottle-feeding among low birth weight preterm; and early progressive feeding compared to delayed feeding among extremely low birth weight preterm. Bovine/cow milk feeding and high volume feeding interventions had an unfavourable effect, while electric breast pump and Galactagogue had a mixed effect. Regarding anthropometric outcomes, overall, macronutrient fortified formula, cream supplementation, and fortified human milk formula had a positive effect (weight gain) on preterm infants. Interventions comparing human breastmilk/donor milk with formula had mixed effects. Overall, only human milk compared to formula intervention had a positive effect on morbidity among preterm infants, while none of the interventions had any positive effect on mortality. Bovine/cow milk supplementation had unfavourable effects on both morbidity and mortality. (4) Conclusion: Future research should prioritise low- and middle-income countries, include infants presenting with growth failure in the post-neonatal period and record effects on morbidity and mortality outcomes.

## 1. Introduction

Early life malnutrition and growth faltering is an important global public health problem [1]. Previously thought to be uncommon, estimates indicate that some 8.5 million infants aged under six months (<6 m) worldwide are wasted (have low weight-for-length, an important anthropometric marker of nutritional risk) [2]. Infants <6 m are not simply small-children; the first six months of life represents a period of rapid maturation and development with unique dietary needs, since infants should ideally be breastfed during this period [3]. The mother or maternal substitute, therefore, plays a critical role in fulfilling the nutritional requirements [4]. Unmet nutritional requirements can have serious implications for growth and survival. The short-term implications include a higher risk of morbidity and mortality, while long-term effects have implications for later health and well-being including the risk of non-communicable diseases [3].

The World Health Organisation (WHO)/United Nations Children’s Fund (UNICEF) global strategy for infant and young child feeding recognises the importance of early initiation of breastfeeding and exclusive breastfeeding (EBF) [5]. However, only 37% of infants <6 m are exclusively breastfed in low- and middle-income countries (LMICs) [6]. Moreover, it is recognised that a significant number of breastfeeding mothers of infants <6 m might face challenges in breastfeeding. Globally, an estimated 15–20% of all births are low birth weight (LBW) [7]. Regarding feeding practices, these infants are often not breastfed and many times not fed at all during the initial hours and days of life [8]. Risk factors for growth failure include both infant and maternal factors. In addition to LBW, sub-optimal feeding practices, congenital abnormalities, and underlying morbidities are common risk factors, while maternal physical and mental conditions are increasingly recognised as other potential causes [9,10]

Many early-life interventions to improve growth among young infants have been tested, ranging from cup feeding to spoon-feeding to the fortification of either human/donor breastmilk or formula [11,12,13,14,15,16,17]. However, currently, especially for infants who are already small or at-risk, there is insufficient data to develop recommendations [14,15,16,18,19,20,21]. In 2011 and 2013, WHO published recommendations on the feeding of LBW infants and management of severe acute malnutrition among infants <6 m, respectively; these recommendations were based on a limited and low or very low quality of evidence [8,22]. Given the importance of early infant feeding practices on morbidity, mortality, and long-term health and well-being, generating high-quality evidence is essential to inform prevention and management of growth failure among young infants.

Through this review, we aim to inform research priorities to prevent and manage growth failure among small and at-risk infants <6 m. The objectives include the following:(1)to identify and describe feeding interventions with a focus on restoring or improving the volume and quality of breastmilk and breastfeeding when breastfeeding practices are sub-optimal or prematurely stopped, and(2)to assess the impact of these interventions on feeding practices, anthropometry, morbidity, and mortality status.

## 2. Materials and Methods

We developed and followed a standard systematic review protocol in accordance with the PRISMA (preferred reporting items for systematic review and meta-analysis protocols) statement [23].

### 2.1. Eligibility Criteria

Population: We reviewed studies involving infants <6 m who are small or at nutrition-related risk, including those with LBW, and those with weight loss or feeding difficulties.

Intervention: Studies were eligible if they focused on feeding interventions for infants, mothers, or both that aimed at restoring or improving the quality and volume of breastmilk and optimising breastfeeding when breastfeeding practices are sub-optimal or prematurely stopped.

Comparison: Studies reporting any comparison between/with interventions of interest.

Outcomes: Studies reporting on at least one of the following outcomes—feeding practices, anthropometry, morbidity, or mortality—were included.

Study design: We selected studies that included randomised control trials, quasi-experimental, cohort, cross-sectional, and other comparative observational studies.

Context: Studies from both LMICs and high-income countries (HICs) were included.

Exclusion criteria included (1) studies with medical interventions, such as use of antibiotics and micronutrients in addition to human/donor/formula milk fortification; (2) unpublished studies; (3) reviews/systematic reviews; (4) non-human studies; (5) studies not published in English; and (6) studies published in abstract form only, correspondence, letters, case studies, opinion pieces, and protocols.

### 2.2. Search Strategy

Searches were conducted in three databases: PubMed, CINAHL Plus, and Cochrane Library. We used the following search strategy for PubMed: (“infant” or “newborn” or “newborns” or “neonate” or “neonates” or “low birth weight” or “low birthweight” or “low-birth-weight” or “LBW” or “small for gestational age” or “small-for-gestational-age” or “SGA” or “premature” or “pre-mature” or “preterm” or “pre-term” or “severe acute malnutrition” or “malnutrition” or “SAM” or “wasting” or “wasted” or “foetal growth retardation” or “foetal growth restriction” or “fetal growth restriction” or “fetal growth retardation” or “intrauterine growth restriction” or “intrauterine growth retardation” or “IUGR” or “failure to thrive” or “FTT” or “growth failure” or “growth faltering”) and (“human milk” or “breast milk” or “breastmilk” or “infant formula” or “establishing breastfeeding” or “supplement*” “suck*” or “spoon fe*” or “cup fe*” or “bottle feeding” or “breast milk substitute” or “breast milk fortifier” or “infant feeding practices” or “early weaning” or “relact*” or “complementary food” or “supplementary food”). Similar keywords were used with other selected databases. We limited the evidence to studies published in the English language from Jan 1990 to December 2018 and focusing on the human species.

### 2.3. Study Selection

All identified records were imported in Eppi Reviewer software (version V.4.8.0.0, EPPI-Centre, UCL Institute of Education, University of London, London, UK). Using a two-stage screening process, two reviewers independently screened all titles and abstracts (first stage), and full texts (second stage); any disagreements were resolved by a third reviewer.

### 2.4. Data Extraction

Two reviewers independently extracted data using standard data extraction codes developed for this study. A third reviewer checked the coding in Eppi Reviewer. We extracted data on population (including sample size, details of setting, and country), intervention (description), comparison, outcome (description, type of measurement, effect size, and strength of evidence), and study design.

### 2.5. Analysis

Since the review examines a range of interventions and outcomes, the analyses are presented as a narrative synthesis. However, where the authors had given the magnitude of effect (including statistical uncertainty using confidence intervals) and strength of evidence, it is presented in the results table. We used the following terms for direction of effect: (1) positive evidence of uniformly favourable impacts across one or more outcome measures, analytic samples (full sample or subgroups), and/or studies; (2) negative evidence of uniformly adverse impacts across one or more outcome measures, analytic samples (full sample or subgroups), and/or studies; (3) no effect evidence of uniformly null impacts across one or more outcome measures, analytic samples (full sample or subgroups), and/or studies; and (4) mixed effect evidence of a mix of favourable, null, and/or adverse impacts across one or more outcome measures, analytic samples (full sample or subgroups), and/or studies.

## 3. Results

### 3.1. Study Selection

Figure 1 presents the selection process and search results. The search identified 16,638 records. After duplicate removal and initial screening of titles and abstracts, 177 records were eligible for full-text review. Among them, 130 did not meet the inclusion criteria — 107 studies did not focus on targeted interventions, 13 studies did not report outcomes of interest, seven studies were published as protocol/abstract form only/correspondence/opinion, two studies did not focus on the targeted population, and one study was a duplicate. Finally, 47 studies were included in the analysis.

### 3.2. General Characteristics of the Included Studies

Table 1 presents a summary of the descriptive characteristics of the included studies. HICs represented three fourths of the studies (*n* = 35, 74.5%), with the highest number of studies from the USA (*n* = 21, 44.7%). Most studies were randomised control trials (RCT) (*n* = 38, 80.8%), and the sample size of studies ranged from 20 to 642. Regarding population focus, the majority included preterm with LBW/very low birthweight (VLBW)/extremely low birthweight (ELBW) (*n* = 41, 87.2%), while a few included mothers of preterm (*n* = 3, 6.4%) and infants with faltering growth (*n* = 3, 6.4%). We categorised identified interventions into the following groups: formula fortification or supplementation (*n* = 31, 66%), enteral feeds (*n* = 8, 17%), cup feeding (*n* = 2, 4.3%), and other interventions (*n* = 6, 12.7%). Similarly, identified outcomes were categorised as anthropometry (*n* = 40, 85.1%), feeding (*n* = 16, 34%), morbidity (*n* = 11, 23.4%), and mortality (*n* = 5, 10.6%).

Table 2 presents a summary of included reviews. A more detailed summary (intervention components and outcome measures) is presented in Appendix B (Table A1). The subsequent section briefly describes the effect of included studies.

### 3.3. Synthesis of Results

#### 3.3.1. Cup Feeding

Two studies compared cup feeding with bottle-feeding among preterm infants [24,25]. Abouelfettoh et al. found a higher proportion of infants being breastfed one week post-discharge in the cup feeding group (80% vs. 64%, *p* = 0.03). Yilmaz et al. also found a significantly higher proportion of infants being exclusively breastfed (at discharge (72 vs. 46, *p* < 0.0001), 3 m (77 vs. 47, *p* < 0.0001), and 6 m (57 vs. 42, *p* < 0.001)) in the cup feeding group. This study also measured weight gain during the first seven days; however, no difference was observed between cup feeding and bottle-feeding (a bottle with a teat or nipple with formula or breast milk).

#### 3.3.2. Formula Fortification or Supplementation

##### Bovine/Cow Milk Based Formula

Two studies compared bovine/cow milk based formula with fortified human milk. Abrams et al. assessed the effect of a diet consisting of either human milk fortified with a human milk protein-based fortifier (HM) or a diet containing variable amounts of milk containing cow milk-based protein (CM) among VLBW preterm infants [26]. No differences were observed in weight and length change during neonatal intensive care unit stay. Regarding morbidity, the CM group had a higher number of necrotizing enterocolitis (NEC) cases than the HM group (17% vs. 5%, *p* = 0.002); however, a similar significant effect was not observed in the case of sepsis. In addition, the CM group also had higher mortality compared to the HM group (8% vs. 2%, *p* = 0.04). Cristofalo et al. compared the effect of bovine milk-based preterm formula with HM fortification [33]. ELBW preterm infants fed with fortified bovine milk had a higher duration of parental nutrition (36 vs. 27 days, *p* = 0.04) and more cases of NEC (5 vs. 1, *p* = 0.08) compared to infants fed with HM based formula. However, no significant effect was observed on mortality between the two groups.

##### Protein Supplementation

Four studies evaluated the effect of protein supplementation on anthropometry [27,29,36,40]. Two studies compared human milk (HM) supplemented with extra protein to HM with standard fortification; the authors reported a mixed effect on anthropometric outcomes [27,29]. Another two studies compared (1) partially hydrolysed whey protein with non-hydrolysed whey casein formula [36], and (2) liquid extensively hydrolysed protein with powdered formula [40]. None of the studies found a statistically significant effect on weight or length gain.

##### Lactase Fortification

Two studies assessed the effect of lactase fortification on anthropometry [34,37]. Erasmus et al. focused on preterm, while Gathwala et al. studied term small for gestational age (SGA) infants. Erasmus et al. reported no difference in anthropometry at the fourth week between the lactase (Lactaid drops) treated group and untreated fortified HM or preterm formula. In contrast, the study among term SGA infants supplemented with HM fortified with lactase (Lactodex) reported an improvement in weight gain (38.7 vs. 28.7 g/d, *p* < 0.001) and length gain (1.14 vs. 0.87 cm/wk, *p* < 0.01) at the fourth week [37].

##### Fortification with Iron

Two studies compared human milk fortified with iron versus standard fortification among preterm infants [55,70]. Both studies reported no effect on anthropometric outcomes. Willeitner et al. also measured morbidity and mortality outcomes [55]: again, no difference was observed between the intervention and control groups.

##### Nutrient Fortification

Three studies evaluated the effect of nutrient fortification [32,46,56]. Clarke et al. compared the effect of nutrient-dense formula with the energy-supplemented formula on weight and length gain among infants with growth faltering (range: 2–31 wk) [32]. Another group studied the effect of nutrient-fortified formula (higher protein, calcium, phosphorous, and other vitamins and minerals) with standard formula among VLBW infants [56]. Both studies reported no effect on anthropometric outcomes. In contrast, Morlacchi et al. observed a significant improvement in weight (205.5 vs. 155 g/wk, *p* = 0.025) and length (1.6 vs. 1.1 cm/wk, *p* = 0.003) among VLBW preterm infants. Here, the authors compared macronutrient fortification of formula with standard fortification [46].

##### Cream Supplementation

Hair et al. assessed the effect of HM-derived cream supplement on LBW infants; the other group consisted of mothers’ own milk or donors’ HM derived fortifier [38]. The cream supplemented group had a significantly higher weight (14.0 vs. 12.4, g/kg/d, *p* = 0.03) and length velocity (1.03 vs. 0.83, cm/wk, *p* = 0.02) measured at 36 wks post-menstrual age or weaned from fortifier.

##### Early and Delayed Fortification

Three studies assessed the effect of early and delayed fortification [53,54,71]. Shah et al. compared 20 mL/kg/d HM feeds with 100 mL/kg/d feeds [71], while Tillman et al. compared early (1st feeding) fortification with 50–80 mL/kg/d feeds [54], and Taheri et al. compared early (1st feeding) fortification with 75 mL/kg/d feeds [53]. None of these reported any effect on anthropometry. Shah et al. and Taheri et al. also measured the effect on feeding outcomes; however, none found a significant difference between early and delayed fortification groups. Taheri et al. also recorded morbidity and observed no differences.

##### Fortified Human Milk

Two studies evaluated the effect of HM fortification on VLBW infants [31,47]. Bhat et al. compared fortified HM with only human milk; the authors reported better weight gain among the intervention group at two months [31]. Morlacchi et al. compared fortified HM with preterm formula; the authors did not observe any difference in weight and length between the two groups at term corrected age [47]. However, infants fed with fortified HM had less fat mass (14.9% vs. 19.2%, *p* = 0.002) and more fat-free mass (85.1% vs. 80.8%, *p* = 0.002) compared to the formula-fed group.

##### Different Formulas

Five studies compared the effect of HM (mother’s own or donor) and formula [41,43,45,48,50]. Three studies focused on ELBW preterm [41,45,50]. Two studies did not observe any effect on anthropometry [41,50], while Manea et al. reported an improvement in weight among infants fed with human breastmilk (120.8 vs. 97.2 g/wk) [45]. Manea et al. and O’Connor et al. also reported lower morbidity among the intervention group [45,50]. Another two studies focused on LBW and VLBW preterm infants [43,48]. Both studies reported no difference in weight and length between intervention and control groups.

One study assessed the effect of liquid and powdered HM fortification [49]. Moya et al. reported a higher weight (1770 vs. 1670 g, *p* = 0.038) and length gain (41.8 vs. 40.9 cm, *p* = 0.010) among ELBW preterm fed with liquid HM fortification. However, no difference was observed in morbidity between the two groups.

One study compared different levels of fortification (human milk fortifier-HMF) standard (1.2 g HMF + 30 mL HM), moderate (1.2 g HMF + 25 mL HM), and aggressive (1.2 g HMF + 20 mL HM) [39]. The authors reported no significant differences in weight and length gain between the three groups.

Two studies compared the different compositions of fortifiers [42,51]. Porcelli et al. compared Wyeth Nutritional International’s new HMF (test) with Enfamil HMF (reference); the authors reported an improvement in weight gain (19.7 vs. 16.8 g/kg/d, *p* = 0.04); however, no effect was observed on length gain [51]. This study also found a positive effect on feeding outcomes; mean human milk intake was higher in test HMF compared to reference HMF. Similarly, Kumar et al. compared Similac liquid HMF with Enfamil liquid HMF [42]; the authors reported better weight gain among infants fed with Similac liquid HMF.

Another two studies compared different formulas [28,44]. Amesz et al. compared post-discharge formula, term formula, and HM fortified formula [28]; the authors did not find any differences in anthropometric outcomes between the three groups. In contrast, Lucas et al. compared follow-on preterm formula with standard term formula [44]; they reported an improvement in both weight and length gain among infants fed with follow-on preterm formula.

##### Continued EBF and Early Limited Formula

One study compared the effect of early limited formula with continued exclusive breastfeeding among term infants with weight loss within 36 hours of birth [35]. The authors observed a higher proportion of infants being exclusively breastfed at wk 1, 1 m, 2 m, and 3 m among infants fed with early limited formula. Additionally, this group also showed lower weight loss compared to the continued exclusive breastfeeding group (6.8 vs. 8.1%, *p* = 0.10).

#### 3.3.3. Enteral Feed Interventions

One study compared continuous nasogastric gavage (CNG) with intermittent bolus gavage (IBG) among two groups: VLBW and ELBW preterm [57]. The authors reported no difference between CNG and IBG concerning full enteral feeds and regaining birth weight. Similarly, Mosqueda et al. compared intravenous feeds alone with a small bolus of nasogastric feeding among ELBW preterm [61]; the authors observed no effect on either anthropometry or morbidity. Another study compared nasogastric tube feeding with bottle-feeding among LBW preterm [60]; the authors reported higher chances of breastfeeding at discharge, third day, and three months among infants fed with nasogastric tube compared to bottle-fed infants.

Two studies compared different levels of enteral feeds [58,59]. Bora et al. compared complete enteral feeds (CEF) along with expressed breastmilk with minimal enteral feeds (MEF) along with intravenous feeds [58]. Although no difference was reported for feed intolerance and morbidity between the two groups, infants fed with CEF expended fewer days to regain birth weight compared to the MEF group (10.6 vs. 11.8 days, *p* = 0.03). Colaizy et al. compared four levels ( <25%, 25–50%, 50–75%, and >75%) of total enteral intake as human milk, donor milk, or mixed feed [59]. Infants fed with >75% enteral feeds were far below the reference median for weight-for-age z score compared to the other three groups (G4: −0.6 vs. G1, G2, G3: −0.1, −0.3, −0.32, *p* = 0.03).

One study compared high volume feeds (300 mL/kg/d of expressed breastmilk) with standard volume feeds (200 mL/kg/d of expressed breastmilk) [62]. The authors reported a higher number of infants experiencing feed intolerance among the high volume feed group (14 vs. 8, *p* = 0.07). Additionally, this study found a positive effect on weight gain but no effect on morbidity. Similarly, another study compared early progressive feeding (without trophic feeding) with delayed progressive feeding (after trophic feeding) [52]. The authors reported no difference in anthropometry-, morbidity-, and mortality-related outcomes between the two groups. Interestingly, infants in the delayed progressive feeding group reached full enteral feeding in fewer days (17 vs. 19, *p* = 0.02). Another study compared proactive feeding regimen (1st day—100, last day—200 mL/kg/d) with standard regimen (1st day—60, last day—170 mL/kg/d) [63]. The authors reported a significantly better (near to median) change in weight (−0.29 vs. −0.48, *p* = 0.002) and length (−0.19 vs. −0.45, *p* = 0.011) z scores among proactive feeding group.

#### 3.3.4. Other Interventions

One study evaluated the effect of different concentration of bee honey [64]. Compared to control with no honey, other intervention groups (5 g, 10 g, and 15 g honey) demonstrated weight gain by the second week. Two studies assessed the effect of electric breast pump on mothers of preterm infants [65,69]. One study found no difference in volume of breast milk expressed between intervention (electric breast pump + education) and control group (only education) [65], while another study observed a significantly higher volume of breast milk expressed with electric breast pump compared to hand expression (578 vs. 463 mL, *p* < 0.01) [69].

One study compared nasogastric feeds with spoon feeds [66]. The authors observed no difference between the two groups when compared for weight gain. One study assessed the effect of suckling and swallowing exercise [67]. The authors compared two intervention groups (non-nutritive suckling exercise/swallowing exercise) with standard care and reported no difference in start to independent oral feeding between the three groups. One study evaluated the effect of Galactagogue provided to mothers of preterm infants [68]. Compared to placebo, Galactagogue group mothers experienced higher breast milk expression at three months (22 vs. 12, *p* < 0.05); however, this effect was not sustained at six months.

## 4. Discussion

### 4.1. Summary of Key Findings

Overall, interventions with positive effect on feeding practices (most commonly assessed by increased duration of breastfeeding) were cup feeding compared to bottle-feeding among preterm; nasogastric tube feed compared to bottle-feeding among LBW preterm; and early progressive feeding compared to delayed feeding among ELBW preterm. Bovine/cow milk feeding and high volume feeding interventions had an unfavourable effects (feeding intolerance and higher parenteral nutrition days), while electric breast pump and Galactagogue had a mixed effect.

Most of the studies reported anthropometric outcomes. Overall, macronutrient fortified formula, cream supplementation, and fortified human milk formula had positive effects (weight gain) on preterm infants. Interventions comparing human breastmilk/donor milk with formula had mixed effects.

Overall, only human milk compared to formula intervention had a positive effect on morbidity among preterm infants, while none of the interventions had any positive effects on mortality. Moreover, bovine/cow milk supplementation had unfavourable effects on morbidity and mortality, respectively.

### 4.2. This Review’s Findings in Context

Most current evidence is based on limited studies with low to medium study quality [14,15,16,18,19,21]. A review assessing the effect of cup feeding versus other forms of supplemental enteral feeding, for infants unable to fully breastfeed, reported similar findings as observed in our review [11]. However, the authors highlighted the challenge with compliance to cup feeding. Another review examining the effect of formula versus donor human milk for preterm or LBW infants showed better anthropometric outcomes among the formula-fed group [72]. In contrast, our review observed a mixed effect. This difference may be due to the inclusion of all studies in our review, irrespective of study quality. Similar to our findings, a review by Amissah et al. also reported a positive effect on weight with protein supplementation [20]. In decisions around early use of formula milks, it is important to note discussions around risk of allergies that may arise as a result. One recent review focusing on allergies found a risk ratio of 1.75 (95% CI: 1.30–2.27, *p* = 0.0001) for breastfed infants given cow’s milk formula supplementation in the first few weeks of life against no supplementation given to breastfed infants [73].

We conducted this review with an aim to identify research priorities to prevent and manage growth failure in the first six months of life. The included studies reported a range of populations, interventions, outcomes, and contexts. These are discussed further in the following section.

The population included preterm LBW/VLBW/ELBW, term SGA with LBW, mothers of preterm, and newborns with growth faltering in early days. Interestingly, our search identified a majority of those young infants who are at risk of growth failure at birth. This is indicative of a research gap among other infants under six months where growth failure may manifest or present after birth at different life stages (e.g., early growth failure before 12 wks/3 months or later growth failure-after 12 wks/3 months).

Identified interventions included cup feeding; formula fortification or supplementation with macro and micronutrients; lactase supplementation; bovine/cow milk vs. human milk human milk vs. formula; early vs. delayed feeding; low/high vs. standard feeding; nasogastric tube vs. intravenous feeding; suckling/swallowing exercise vs. standard feeding; nutrient-dense vs. standard feeding; and electric breast pump and Galactagogue.

Reported feeding outcomes included any or exclusive breastfeeding, feed intolerance, days on parenteral nutrition, days to oral feeds, and maternal milk volume. Anthropometric outcomes included-weight/length gain, weight/length velocity, weight-for-age/length-for-age z scores, weight loss, fat mass, fat-free mass, birth weight regain, and weight/length centiles. Similarly, morbidity outcomes included sepsis, NEC, and infection.

Lastly, most (75%) of the evidence is from HICs. However, LMICs have a higher prevalence of growth faltering. This highlights the need for designing and testing further interventions in LMIC settings. Similarly, the majority of the interventions tested were based on inpatient tertiary care hospital settings. These may have limited applicability and could even pose a health risk if applied in a low resource setting without suitable skillset, environmental conditions, and infrastructure. Hence, given the limited health care infrastructure and skillset in LMICs [74,75], future interventions should be suited to and explored in community-based settings. Although some work has been initiated in this area [76], much more still needs to be done to prevent and manage growth faltering among young infants. In rural Rwanda, using a medical-home model, integrated care is provided to at-risk infants through paediatric developmental clinics [77]. Similarly, the C-MAMI (community management of at-risk mothers and infants under six months) tool is being tested in Gambella refugee camps in Ethiopia and in the Rohingya response in Bangladesh [78,79]. Other interventions could include breastfeeding support (supplementary suckling, special breastfeeding support for vulnerable infants, support to mothers to increase confidence, supporting adolescent and working mothers) and non-breastfeeding support (partner, group, and community support).

In addition to the effectiveness of identified feeding interventions, we also extracted information on potential biases. Only one-third of the included studies had sample size more than 100 (50 infants in each arm or more), while nearly a third had sample size below 60 (30 infants each arm or less). This is of concern particularly for studies where authors have also conducted sub-group analysis. Additionally, lost to follow up was a concern in most of the studies, although not formally calculated. Similarly, authors did not report compliance to intervention in many studies. One study reported a challenge with the determination of breastfeeding practices beyond six weeks after discharge [24]. The authors highlighted low education level among mothers as a limiting factor for maternal verbal recall. Similarly, another study emphasised the limitation of determining the duration of full and partial breastfeeding around six months of age [60].

### 4.3. Strengths and Limitations

We did not formally assess the study quality of each individual study, but despite there being a good number of RCTs in our final sample, overall quality of studies was not always high, a common challenge being small sample size. Additionally, despite many studies reporting on anthropometric change, it is morbidity and mortality that really matters as a key outcome. Even when this was reported, only short-term outcomes were assessed; long term changes may also be relevant, especially given increasing appreciation of links between early life growth and later life risk of non-communicable diseases [80]. In recent years, there has been significant debate about the susceptibility of research to biases of various kinds, one of which includes industry-funded science [81]. Nearly a quarter of studies reported financial support from industry (Appendix A, Appendix A), while another 19 studies did not declare whether they received any financial support for the study. Our literature search captured studies published until December 2018.

The review findings should be interpreted considering these limitations. The present review also had several strengths, including the broad scope of the review covering a range of interventions and methodological rigor (double data screening and extraction). Furthermore, this review identified research priorities to prevent and manage growth faltering among a group of young infants that were either excluded or missed in earlier research studies.

## 5. Conclusions

This review explored ways to manage the feeding of small and at-risk infants in the first six months of life. Whilst finding a large range of interventions, most studies were set in HICs and focused on infants identified around the time of birth with risk factors such as with LBW. Future research needs to do more in LMICs where not only is the problem more common but the consequences more severe. More focus is also needed on infants who present in the post-neonatal period with growth faltering (either new onset or because earlier risk factors like LBW have not previously been noted or acted upon). Although most of the included studies recorded anthropometric outcomes, future research should also record effects on morbidity and mortality outcomes. Ideally, not just in the short term, but also any longer-term impacts.

## Figures and Tables

**Figure 1 nutrients-12-02044-f001:**
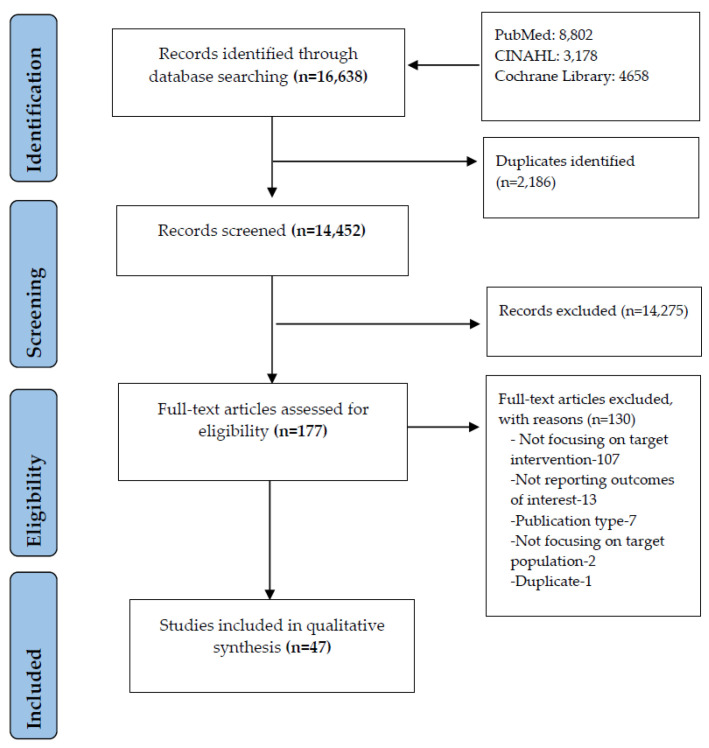
Flow diagram of included and excluded studies.

**Table 1 nutrients-12-02044-t001:** Characteristics of included studies (*n* = 47).

Author (Year)	Country	Study Design	Population	Sample Size *	Outcomes			
					Feeding	Anthropometry	Morbidity	Mortality
**Cup Feeding Interventions (2)**								
Abouelfettoh (2008) [24]	Egypt	QE	Preterm (LBW)	60	✓			
Yilmaz (2014) [25]	Turkey	RCT	Preterm (VLBW)	607	✓	✓		
**Formula Fortification/Supplementation Interventions (31)**								
Abrams (2014) [26]	USA	RCT	Preterm (VLBW)	260		✓	✓	✓
Alan (2013) [27]	Turkey	PO	Preterm (VLBW)	58		✓		
Amesz (2010) [28]	Netherlands	RCT	Preterm (VLBW)	102		✓		
Arslanoglu (2006) [29]	Italy	RCT	Preterm (LBW, VLBW, ELBW)	34		✓		
Berseth (2004) [30]	USA	RCT	Preterm (VLBW)	181		✓		
Bhat (2001) [31]	Oman	RCT	Preterm (VLBW)	100		✓		
Clarke (2007) [32]	USA	RCT	Faltering growth	60		✓		
Cristofalo (2013) [33]	USA	RCT	Preterm (ELBW)	53	✓		✓	✓
Erasmus (2002) [34]	Canada	RCT	Preterm (VLBW)	130		✓		
Flaherman (2013) [35]	USA	RCT	Term (weight loss)	40	✓	✓		
Florendo (2009) [36]	USA	RCT	Preterm (VLBW)	80		✓		
Gathwala (2007) [37]	India	RCT	Term SGA (LBW)	65		✓		
Hair (2014) [38]	USA	RCT	Preterm (ELBW)	78		✓		
Kanmaz (2012) [39]	Turkey	RCT	Preterm (ELBW)	84		✓		
Kim (2015) [40]	USA	RCT	Preterm (VLBW)	147		✓		
Kim (2017) [41]	South Korea	Cohort-R	Preterm (ELBW)	132		✓		
Kumar (2017) [42]	USA	RCT	Preterm (ELBW)	31		✓		
Lok (2017) [43]	Hong Kong	Cohort-R	Preterm (LBW, VLBW)	642		✓		
Lucas (1992) [44]	UK	RCT	Preterm (VLBW)	32		✓		
Manea (2016) [45]	Romania	QE	Preterm (ELBW)	34		✓	✓	
Morlacchi (2016) [46]	Italy	QE	Preterm (VLBW)	20		✓		
Morlacchi (2018) [47]	Italy	PO	Preterm (VLBW)	32		✓		
Morley (2000) [48]	UK	RCT	Preterm (LBW)	96		✓		
Moya (2012) [49]	USA	RCT	Preterm (ELBW)	150		✓	✓	
O’Connor (2016) [50]	Canada	RCT	Preterm (ELBW)	363		✓	✓	✓
Porcelli (1999) [51]	USA	RCT	Preterm (VLBW, ELBW)	64	✓	✓		
Shah (2016) [52]	USA	RCT	Preterm (VLBW)	100	✓	✓		
Taheri (2016) [53]	Iran	RCT	Preterm (VLBW)	72	✓	✓	✓	
Tillman (2012) [54]	USA	Pre-post	Preterm (VLBW)	95		✓		
Willeitner (2017) [55]	USA	RCT	Preterm (VLBW, ELBW)	70		✓	✓	✓
Worrell (2002) [56]	USA	Cohort-R	Preterm (VLBW)	180		✓		
**Enteral Feed Interventions (8)**								
Akintorin (1997) [57]	USA	RCT	Preterm (VLBW, ELBW)	80	✓	✓		
Bora (2017) [58]	India	RCT	Preterm (VLBW)	107	✓	✓	✓	
Colaizy (2012) [59]	USA	RCT	Preterm (ELBW)	171		✓		
Kliethermes (1999) [60]	USA	RCT	Preterm (LBW)	84	✓			
Mosqueda (2008) [61]	USA	RCT	Preterm (ELBW)	84		✓	✓	
Salas (2018) [52]	USA	RCT	Preterm (ELBW)	60	✓	✓	✓	✓
Thomas (2012) [62]	India	RCT	Preterm (VLBW)	61	✓	✓	✓	
Zecca (2014) [63]	Italy	RCT	Preterm (LBW)	72		✓		
**Other Interventions (6)**								
Aly (2017) [64]	Egypt	RCT	Preterm (VLBW)	40		✓		
Heon (2016) [65]	Canada	RCT	Mothers of extremely preterm	40	✓			
Kumar (2010) [66]	India	RCT	Preterm (VLBW)	144		✓		
Lau (2012) [67]	USA	RCT	Preterm (VLBW)	70	✓			
Serrao (2018) [68]	Italy	RCT	Mothers of preterm	100	✓			
Slusher (2007) [69]	Nigeria and Kenya	RCT	Mothers of preterm	65	✓			

Symbol: * participants included in each study. Abbreviations: ELBW = extremely low birthweight, LBW = low birth weight, PO = prospective observational, QE = quasi experimental, R = retrospective, RCT = randomised controlled trial, SGA = small for gestational age, VLBW = very low birth weight.

**Table 2 nutrients-12-02044-t002:** Effect of feeding interventions on feeding practices, anthropometry, morbidity and mortality outcomes (*n* = 47).

Author (Year)	Population Characteristics	Intervention	Outcomes			
			Feeding Practices	Anthropometry	Morbidity	Mortality
		**Cup Feeding Interventions (*n* = 2)**				
Abouelfettoh (2008) [24]	Preterm (LBW) (GA: 35.13 wk, Bwt: 2150 g)	**Cup feeding** IG: Cup feeding vs. CG: Bottle feeding	Positive			
Yilmaz (2014) [25]	Preterm (VLBW) (GA: 32–35 wk, Bwt: 1543 g)	**Cup feeding** G1: Cup feeding vs. G2: Bottle feeding	Positive	No effect		
		**Formula Fortification/Supplementation Interventions (31)**				
Abrams (2014) [26]	Preterm (VLBW) (Bwt: <1250 g)	**Bovine/cow milk** G1: Cow milk (CM formula + CM based fortifier) G2: Human milk (HM (mother’s own/donor milk) + HM based fortifier)		No effect	Negative	Negative
Cristofalo (2013) [33]	Preterm (ELBW) (GA: <27 wk, Bwt: 989 g)	**Bovine/cow milk** G1: Exclusive appropriately fortified HM G2: Bovine milk-based preterm formula	Negative		Negative	No effect
Alan (2013) [27]	Preterm (VLBW) (GA: ≤32 wk, Bwt: ≤1500 g)	**Protein supplementation** IG: HM with extra protein supplementation CG: HM with a standard fortification		Mixed		
Arslanoglu (2006) [29]	Preterm (LBW, VLBW, ELBW) (GA: 26–34 wk, Bwt: 600–1750 g)	**Protein supplementation** G1: HMF (with additional protein) G2: HM with HMF (standard amount)		Mixed		
Florendo (2009) [36]	Preterm (VLBW) (GA: ≤32 wk, Bwt: 1200 g)	**Protein supplementation** IG: Partially hydrolysed whey protein CG: Non-hydrolysed whey casein preterm infant formula		No effect		
Kim (2015) [40]	Preterm (VLBW) (GA: ≤33 wk, Bwt: 1174 g)	**Protein supplementation** IG: Conc. HMF containing liquid extensively hydrolysed protein CG: Powdered intake protein HMF		No effect		
Erasmus (2002) [34]	Preterm (VLBW) (GA: 26–34 wk, Bwt: 1407 g)	**Lactase fortification** IG: Fortified HM or preterm formula treated with lactaid drops (Lactase) CG: Untreated fortified HM or preterm formula		No effect		
Gathwala (2007) [37]	Term SGA (LBW) (GA: 40 wk, Bwt: 2000 g)	**Lactase fortification** IG: HM fortified with Lactodex-HMF vs. CG: Only BM		Positive		
Berseth (2004) [70]	Preterm (VLBW) (GA: ≤33 wk, Bwt: 1180 g)	**Iron fortification** G1: HMF (iron fortified) vs. G2: HMF (standard)		No effect		
Willeitner (2017) [55]	Preterm (VLBW, ELBW) (GA: 29 wk, Bwt: 500–1499 g)	**Iron fortification** IG: HM fortification (Concentrated preterm formula 30 Similac Special Care 30 with iron) CG: Standard Powdered HMF (Similac HMF)		No effect	No effect	No effect
Clarke (2007) [32]	Faltering growth (GA: 2–31 wk)	**Nutrient fortification** G1: Nutrient-dense formula G2: Energy-supplemented formula		No effect		
Morlacchi (2016) [46]	Preterm (VLBW) (GA: <32 wk, Bwt: 1255 g)	**Nutrient fortification** G1: Macronutrient fortification G2: Standardised fortification		Positive		
Worrell (2002) [56]	Preterm (VLBW) (GA: 27 ± 3 wk, Bwt: 925 g)	**Nutrient fortification** G1: Transitional formula (higher amounts of protein, Ca, *p*, and several trace minerals and vitamins) G2: Standard formula		No effect		
Hair (2014) [38]	Preterm (ELBW) (GA: 28 wk, Bwt: 970 g)	**Cream supplementation** IG: HM derived cream supplement CG: Mothers own milk or donor’s HM derived fortifier		Positive		
Shah (2016) [71]	Preterm (VLBW) (GA: 27 wk, Bwt: <1500 g)	**Early and delayed fortification** G1: Early fortification (20 mL/kg/d of HM feeds) G2: Delayed fortification (100 mL/kg/d of HM feeds)	No effect	No effect		
Taheri (2016) [53]	Preterm (VLBW) (GA: 28–34 wk, Bwt: 1294 g)	**Early and delayed fortification** G1: Early fortification (1st feeding) G2: Late fortification (BF volume reached 75 mL/kg/d)	No effect	No effect	No effect	
Tillman (2012) [54]	Preterm (VLBW) (GA: <31 wk, Bwt: 1123 g)	**Early and delayed fortification** Fortification with Enfamil, powdered HM fortifier G1: Early BM fortification (1st feed) G2: Delayed fortification (when volume reached 50–80 mL/kg/d)		No effect		
Bhat (2001) [31]	Preterm (VLBW) (GA: 26–34 wk, Bwt: 1242 g)	**Human milk fortification** IG: Fortified HM vs. CG: HM only		Positive		
Morlacchi (2018) [47]	Preterm (VLBW) (GA: <32 wk, Bwt: <1500 g)	**Human milk fortification and formula** G1: Fortified HM vs. G2: Preterm formula		Mixed		
Kim (2017) [41]	Preterm (ELBW) (GA: <32 wk, Bwt: 1087 g)	**Human milk and formula** G1: Donor human milk vs. G2: Preterm formula		No effect		
Lok (2017) [43]	Preterm (LBW, VLBW) (GA: <37 wk, Bwt: <2200 g, VLBW: <1500 g, LBW: ≥1500 g and <2200 g)	**Human milk and formula** Category 1: LBW, Category 2: VLBW; Both the groups further divided into G1: Any BM (human/donor) vs. G2: No BM (infant formula)		No effect		
Manea (2016) [45]	Preterm (ELBW) (GA: 25–33 wk, Bwt: <1000 g)	**Human milk and formula** G1: Human BM vs. G2: Formula		Positive	Positive	
Morley (2000) [48]	Preterm (LBW) (GA: ≤31 wk, Bwt: <1850 g)	**Human milk and formula** Category 1: As sole diet, Category 2: As supplement to HM G1: Banked donor milk vs. G2: Preterm formula		No effect		
O’Connor (2016) [50]	Preterm (ELBW) (GA: 27.5 wk, Bwt: 995 g)	**Human milk and formula** G1: Donor milk vs. G2: Preterm formula		No effect	Positive	No effect
Moya (2012) [49]	Preterm (ELBW) (GA: ≤30 wk, Bwt: 1000 g)	**Liquid and powdered fortification** G1: Liquid HMF vs. G2: Powdered HMF		Positive	No effect	
Kanmaz (2012) [39]	Preterm (ELBW) (GA: 28 wk, Bwt: 1092 g)	**Different levels of fortification** G1: Standard fortification (1.2 g HMF + 30 mL HM) G2: Moderate fortification (1.2 g HMF + 25 mL HM) G3: Aggressive fortification (1.2 g HMF + 20 mL HM)		No effect		
Porcelli (1999) [51]	Preterm (VLBW, ELBW) (GA: 25–32 wk, Bwt: 600–1500 g)	**Different fortifier** G1: Test HMF (1 g of protein/100 mL of supplemented milk, 85% glucose polymers, 15% lactose, and calcium, phosphorus, sodium, copper) G2: Reference HMF (60% whey protein and 40% casein, 75% glucose polymers, 25% lactose and calcium, phosphorus, sodium, and copper)	Positive	Mixed		
Kumar (2017) [42]	Preterm (ELBW) (GA: 27 wk, Bwt: 993 g)	**Different formula** G1: Similac liquid HMF G2: Enfamil liquid HMF		Positive		
Amesz (2010) [28]	Preterm (VLBW) (GA: ≤32 wk, Bwt: 1338 g)	**Different formulas** G1: Post discharge formula G2: Term formula G3: HM fortified formula		No effect		
Lucas (1992) [44]	Preterm (VLBW) (GA: 31 wk, Bwt: 1475 g)	**Different formula** G1: Follow-on preterm formula G2: Standard term formula		Positive		
Flaherman (2013) [35]	Term infants ( >37 wk who lost ≥5% Bwt before 36 h of age)	**Continued EBF and early limited formula** IG: Early limited formula (10 mL using feeding syringe) CG: Continued EBF	Positive	Positive		
		**Enteral feed Interventions (8)**				
Akintorin (1997) [57]	Preterm (VLBW, ELBW) (GA: 28 wk, Bwt: 700–1250 g) Category 1: 700–1000 g Category 2: 1001–1250 g	**Continuous nasogastric gavage(CNG) and intermittent bolus gavage (IBG) feeds** G1: CNG vs. IBG G2: CNG vs. IBG	No effect	No effect		
Mosqueda (2008) [61]	Preterm (ELBW) (GA: 26 wk, Bwt:760 g)	**Intravenous and nasogastric feeds** G1: Intravenous alimentation alone (NPO (none per orem)) G2: Small boluses of nasogastric feedings		No effect	No effect	
Kliethermes (1999) [60]	Preterm (LBW) (GA: ≤32 wk, Bwt: 1685 g)	**Nasogastric and bottle feeds** G1: Nasogastric tube vs. G2: Bottle feeding	Positive			
Bora (2017) [58]	Preterm (VLBW) (GA: 35 wk, Bwt: 1357 g)	**Complete and minimal feeds** G1: Complete enteral feed (CEF) with EBM G2: Minimal enteral feed (MEF) with IVF	No effect	Positive	No effect	
Colaizy (2012) [59]	Preterm (ELBW) (GA: 27 wk, Bwt: 889 g)	**Different levels of feeds** G1: <25%, G2: 25–50%, G3: 50–75% vs. G4: >75%		Positive		
Thomas (2012) [62]	Preterm (VLBW) (GA: 31.7 wk, Bwt: 1220 g)	**High and standard volume feeds** G1: High volume feeds (300 mL/kg/d of EBM) G2: Standard volume feeds (200 mL/kg/d of EBM)	Negative	Positive	No effect	
Salas (2018) [52]	Preterm (ELBW) (GA: 22–28 wk, Bwt: 833 g)	**Early and delayed feeding** G1: Early progressive feeding without trophic feeding G2: Delayed progressive feeding after 4 d course of trophic feeding	Positive	No effect	No effect	No effect
Zecca (2014) [63]	Preterm (LBW) (GA: 32–36 wk, Bwt: >1499 g)	**Proactive and standard feeds** G1: Proactive Feeding Regimen G2: Standard Feeding Regimen		Positive		
		**Other Interventions (*n* = 6)**				
Aly (2017) [64]	Preterm (VLBW) (GA: ≤34 wk, Bwt: 1300 g)	**Bee honey** G1: 5 g, G2: 10 g, G3: 15 g vs. G4: 0 g (control)		Positive		
Heon (2016) [65]	Mothers of extremely preterm infants	**Electric breast pump** IG: Standard care + double electric breast pump + BM expression education and support intervention CG: Education and support	No effect			
Slusher (2007) [69]	Mothers of preterm (GA: 31 wk)	**Electric breast pump** G1: Electric breast pump G2: Non-electric pedal Pump G3: Hand expression	Mixed			
Kumar (2010) [66]	Preterm (VLBW) (GA: ≥32 wk, Bwt: >1250 ≤ 1600 g)	**Nasogastric and spoon feeds** Trial 1 G1: NG feeding in hospital vs. G2: Spoon feeding in hospital Trial 2 G1: Spoon feeding in hospital vs. G2: Spoon feeding at home		No effect		
Lau (2012) [67]	Preterm (VLBW) (GA: 28 wk, Bwt: 1103 g)	**Suckling and swallowing** IG1: Non-nutritive sucking exercise (pacifier use) IG2: Swallowing exercise (placing a milk/formula bolus through syringe) CG: Standard care	No effect			
Serrao (2018) [68]	Mothers of preterm (GA: 27–32 wk)	**Galactagogue** G1: Silymarin-phosphatidylserine and galega (a daily single dose of 5 g of Piu`latte Plus MILTE) G2: Placebo (a daily single dose of 5 g of lactose)	Mixed			

Notes: No effect: Evidence of uniformly null impacts across one or more outcome measures, analytic samples (full sample or subgroups), and/or studies. Positive: Evidence of uniformly favourable impacts across one or more outcome measures, analytic samples (full sample or subgroups), and/or studies. Negative: Evidence of uniformly adverse impacts across one or more outcome measures, analytic samples (full sample or subgroups), and/or studies. Mixed: Evidence of a mix of favourable, null, and/or adverse impacts across one or more outcome measures, analytic samples (full sample or subgroups), and/or studies. Abbreviations: BM = breastmilk, Bwt = birth weight, CG = control group, EBF = exclusive breastfeeding, EBM = expressed breastmilk, ELBW = extremely low birth weight, GA = gestational age, HM = human milk, HMF = human milk fortifier, IG = intervention group, LBW = low birth weight, VLBW = very low birth weight.

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
