# Peer review of "Feeding Interventions for Infants with Growth Failure in the First Six Months of Life: A Systematic Review"

_nutrients, 2020, doi:10.3390/nu12072044_

Round 1

Reviewer 1 Report

The authors have performed a systematic review and discussed the impact of feeding interventions for infants at risk of growth failure in the first six months of life.

Major comments:

  1. Authors should explicitly explain why 130 studies were excluded. The reasoning provided in the figure1 is not clear and should be detailed in the text.
  2. Were there studies discussing micronutrient supplementation other than iron (such as vitamin A, vitamin D, zinc) considered in this review. If not, why? Similarly, were there studies discussing prebiotic, probiotic, and human milk oligosaccharides supplementation (other than the study on bee honey)?
  3. Does any study report head circumference? Does any study report neurological outcome, and length of stay?
  4. Also, in the morbidity analysis – other than NEC, and sepsis – do studies report other morbidities such as feeding intolerance etc.? 

Minor comments:

  1. The title “Feeding interventions for infants with growth failure in the first six months of life: A systematic review” can be changed to “Feeding interventions for infants with and at risk of growth failure in the first six months of life: A systematic review”
  2. Page 3 – “bovine milk and cow milk supplementation” – Please clarify. Also, do authors mean to refer to cow-milk based formula supplementation? 

Author Response

Thanks

Reviewer 2 Report

The systematic review by Rana et al. examined feeding interventions that aimed to restore or improve the volume or quality of breastmilk or optimizing breastfeeding when breastfeeding practices are sub-optimal or prematurely stopped. The key strength of this study is the broad scope of the review covering a wide range of settings, interventions, and outcomes. The main limitation of this study was the lack of any formal risk of bias assessment for individual studies or assessment of quality of evidence for the body of evidence. The manuscript is well written. My specific questions, comments, and suggestions are provided below.

  1. The Results section of the Abstract enumerates the total number of studies and the number of studies broken down by intervention and outcome. I would also suggest narratively summarizing the key findings (perhaps from section 4.1) from the review in the Results sub-section to facilitate the transition into the Conclusion section where future research directions are suggested.
  2. Line 79: If the protocol of this systematic review has been registered in relevant registries (e.g., PROSPERO), please make sure to mention and provide the registration ID.
  3. Line 82: Suggest following the PICO framework and add the inclusion criterion for control/comparison. Based on Table 2, a wide range of control groups was included. This can also help clarify that pre-post design was eligible (based on Table 1), which is not immediately apparent.
  4. Line 83: Suggest changing “at small and at nutrition-related risk” to “small or at nutrition-related risk.”
  5. Line 85: Suggest clarifying whether the eligible participants receiving the interventions were infants, mothers, or both. While this is apparent for the three specified groups of interventions, the scope of "other interventions" is not immediately apparent and will benefit from this clarification.
  6. Line 106: Studies published before December 2018 were captured by the literature search. December 2018 is relatively far, so this may need to be included as a limitation of the study.
  7. No risk of bias assessment
  8. In the box for the reasons of full-text exclusion in Figure 1, suggest adding the word "wrong" to the reasons ("wrong interventions," "wrong target population," etc.). Please also clarify what "evidence" refers to and why seven studies what excluded due to wrong "evidence."
  9. Suggest changing the country name "Korea" to "South Korea" to avoid any potential confusion.
  10. Please clarify whether the sample size in Table 1 represents the total number of infants included in each study or the number of infants receiving the intervention.
  11. In Table 2, when each potential effect was labeled as “no effect,” “positive,” “negative,” or “mixed,” did the authors consider statistical significance or only the point estimates? This issue is alluded to in the footnotes of Table A1 but does not appear to be directly addressed in the main text.
  12. The use of p-values to determine the strength of evidence (Table A1) seems questionable to me. Please explain the rationale and why the well-established GRADE approach was not used.
  13. Most of the included studies used formula fortification or supplementation as the intervention. The interventions and comparison appear highly heterogeneous, which makes the sub-section 3.3.2. a bit hard to follow. I suggest further breaking down the sub-section 3.3.2 by the specific intervention (human milk with protein supplementation, human milk with lactase fortification, infant formula, etc.) to make it more structured.
  14. Suggest rewriting the following sentences as they have been hard to follow to me: “Interestingly, our search identified a majority of those young infants who are at risk of growth failure at birth. This is indicative of a research gap among other young infant groups where growth failure may manifest or present earlier (e.g. before 12 wks) or later (after 12 wks).”
  15. “In addition to the effectiveness of identified feeding interventions, we also extracted information on potential biases.” Please clarify what other information relevant to risk of bias was extracted besides sample size, and where was the information presented for each study. Also, if sufficient information on potential biases was extracted, it is unclear why the authors chose not to do formal RoB assessments for each included study.
  16. “The present review also had several strengths, including the broad scope of the review covering a range of interventions and the methodological rigor (double data screening and extraction).” The independent screening and extraction by two reviewers is the current standard for systematic reviews. Any systematic reviews without two independent reviewers are prone to error and bias. Therefore, I would not necessarily consider this a strength of this study.
  17. I suggest rewriting the last sentence of the paper: “Ideally, not just in the short term, but also any longer-term impacts.”

Author Response

Thanks

Reviewer 3 Report

Feeding interventions for infants with growth failure 2 in the first six months of life: A systematic review

  • Abstract: Suggest mention that you are including studies in all settings in search strategy, as it may be assumed you are focused only on low and middle income countries.
  • Introduction is thorough and outlines key issues. However it would be better constructed as 3-4 main paragraphs rather than multiple shorter sections.
  • The title and aim of the review need realigning. The stated aim refers to “a focus on restoring or improving the volume and quality of breastmilk and breastfeeding”, yet the title does not mention breastfeeding. Suggest reword the title.
  • Line 109: “Two reviewers independently screened all titles and abstract, and full text”. Presumably this was a two stage process? Title and abstracts screened. Next full text articles of identified studies retrieved, then these were examined: please clarify.
  • Please provide further details of your methods to explain how you ensured the rigour of your search: Did you hand search reference lists of articles retrieved, or contact authors for further details? Did you search grey literature?
  • Flow diagram: The writing in the blue boxes is not visible.
  • Flow diagram: Box outlining reasons for full text exclusion – what is meant by “evidence”?
  • Results and discussion– the following terms are used: “bovine” “cow’s milk” “formula” throughout. Please be consistent in use of terms – infant formula is generally derived from cow’s milk, so I’m unsure why different terms are used to distinguish between “bovine” and “cow’s milk”?

Author Response

Thanks

Round 2

Reviewer 1 Report

Authors have convincingly answered my comments/questions and the revision is much better.

Reviewer 2 Report

The authors have adequately addressed my questions and suggestions. I do not have further feedback on this paper.